



# Measurement report: Characterization and source apportionment of coarse particulate matter in Hong Kong: Insights into the constituents of unidentified mass and source origins in a coastal city in southern China

Yee Ka Wong[1,*], Kin Man Liu[2], Claisen Yeung[2], Kenneth K. M. Leung[3], Jian Zhen Yu[1,4,*]

[1]Division of Environment and Sustainability, Hong Kong University of Science and Technology, Clear Water Bay, Kowloon, Hong Kong
[2]Environmental Central Facility, Hong Kong University of Science and Technology, Clear Water Bay, Kowloon, Hong Kong
[3]Hong Kong Environmental Protection Department, 15/F, East Wing, Central Government Offices, 2 Tim Mei Avenue, Tamar, 10  Hong Kong
[4]Department of Chemistry, Hong Kong University of Science and Technology, Clear Water Bay, Kowloon, Hong Kong

Correspondence to: Yee Ka Wong (envrykwong@ust.hk); Jian Zhen Yu (jian.yu@ust.hk)

**Abstract.** Coarse particulate matter (i.e., PM with aerodynamic diameter between 2.5 and 10 micrometers or $PM_{coarse}$) has been increasingly recognized of its importance in $PM_{10}$ regulation because of its growing proportion in $PM_{10}$ and the

accumulative evidence for its adverse health impact. In this work, we present comprehensive $PM_{coarse}$ speciation results obtained through a one-year long (January 2020–February 2021) joint $PM_{10}$ and $PM_{2.5}$ chemical speciation study in Hong Kong, a coastal and highly urbanized city in southern China. The annual average concentration of $PM_{coarse}$ is $14.9\pm8.6$ μg m$^{-3}$ (±standard deviation), accounting for 45 % of $PM_{10}$ ($32.9\pm18.5$ μg m$^{-3}$). The measured chemical components explain ~75 % of the $PM_{coarse}$ mass. The unexplained part is contributed by unmeasured geological components and residue liquid water

content, supported by analyses by positive matrix factorization (PMF) and the thermodynamic equilibrium model ISORROPIA II. The $PM_{coarse}$ mass is apportioned to four sources resolved by PMF, namely soil dust, copper-rich dust, fresh sea salt, and an aged sea salt factor containing secondary inorganic aerosols (mostly nitrate). Back-trajectory cluster analysis reveals significant variations in source contributions with the air mass origin. Under the influence of marine air mass, $PM_{coarse}$ is the lowest (average = 8.0 μg m$^{-3}$) and sea salt is the largest contributor (47 %), followed by the two dust factors (38 % in total).

When the site receives air mass from the northern continental region, $PM_{coarse}$ increased substantially to 21.2 μg m$^{-3}$, with the two dust factors contributing 90 % of the aerosol mass. The potential dust source areas are mapped using the Concentration-Weighted Trajectory technique, showing either the Greater Bay Area or the greater part of southern China as the origin of fugitive dust emissions leading to elevated ambient $PM_{coarse}$ loadings in Hong Kong. This study, first of this kind in our region, provides highly relevant guidance to other locations with similar monitoring needs. Additionally, the study findings point to

the needs for further research on the sources, transport, aerosol processes, and health effects of $PM_{coarse}$.



# 1 Introduction

Coarse particulate matter (PM$_{coarse}$), defined as PM with aerodynamic diameter of 2.5–10 μm in the World Health Organization's air quality guidelines, play important roles in air quality, public health, and global climate. Progress in reducing fine PM (PM$_{2.5}$) pollution in the past makes it increasingly important to explore possibilities to control PM$_{coarse}$ for PM$_{10}$ regulation. In the United States, PM$_{coarse}$ constitutes half of PM$_{10}$ mass nationwide in 2012–2016 (Hand et al., 2019). The relative contribution of PM$_{coarse}$ to PM$_{10}$ mass was reported to increase by 0.7–1.2 % annually over 2000–2016. While the health impact of PM$_{coarse}$ examined by earlier epidemiological studies were inconclusive (Adar et al., 2014), more recent epidemiological studies in China showed evidence for the adverse health impact of PM$_{coarse}$ (Chen et al., 2019; Lei et al., 2022). The health impact of PM$_{coarse}$ depends on the exposure to and concentration and composition of PM$_{coarse}$, which may explain the varied health implications found in different studies (Adar et al., 2014; Chen et al., 2019; Lei et al., 2022).

Understanding the sources of PM$_{coarse}$ is important for developing control strategies. PM$_{coarse}$ is primarily generated by mechanical processes such as wind and erosion, and the sources can be naturally and anthropogenically related. The natural processes include ejection of sea spray, resuspension of soil dust, and release of plant-related particles, etc. Common anthropogenic PM$_{coarse}$ sources include road dust resuspended by road traffic, brake/tire wearing, construction dust, fly ash and metallurgical process. While PM$_{coarse}$ are mostly directly emitted, certain components in PM$_{coarse}$ can be related to secondary formation. For example, nitrate in the coarse mode is formed by the reaction between nitric acid (HNO$_3$) from oxidation of NO$_x$ and preexisting alkaline aerosols (e.g., sea salt and dust). A recent study showed that mineral dust can serve as a medium for rapid secondary inorganic and organic aerosol formation under high photochemical activity and relative humidity conditions, which has important implications to the life cycle of secondary aerosols (Xu et al., 2020). PM$_{coarse}$ also exerts an impact on earth's climate because of its continuous loading in the atmosphere and its ability to scatter and absorb radiation or act as cloud condensation and ice nuclei (USEPA, 2019).

As a coastal and highly urbanized city and being a part of the Guangdong–Hong Kong–Macao Greater Bay Area (GBA) economic and business hub in southern China, Hong Kong is facing atmospheric PM pollution originated from both local and regional influence. Continuous improvement in local and regional PM concentrations is noted in the last few years (HKEPD, 2020). The ambient PM$_{10}$ concentration has been reduced by 24 % from 42 μg m$^{-3}$ in 2012 to 32 μg m$^{-3}$ in 2019. The reduction was contributed mostly by PM$_{2.5}$, which correspondingly decreased by 32 % from 28 to 19 μg m$^{-3}$. By taking the difference between PM$_{10}$ and PM$_{2.5}$, it can be deduced that PM$_{coarse}$ only decreased slightly from 14 to 13 μg m$^{-3}$ in the corresponding period. Because of the disproportionate reduction in PM$_{2.5}$, the relative contribution of PM$_{coarse}$ to PM$_{10}$ increased from 33 % in 2012 to 41 % in 2019. The analysis has two important implications. First, PM$_{2.5}$ and PM$_{coarse}$ in Hong Kong have different sources. Second, it is important to characterize the sources of PM$_{coarse}$, which has gained increasing importance in PM$_{10}$ contribution.

Previous PM$_{coarse}$ studies in Hong Kong were focused on suburban coastal area (Cohen et al., 2004), roadside environment (Cheng et al., 2015), and public transport micro-environments (Jiang et al., 2017). These studies provide limited representation


of the general PM$_{coarse}$ pollution characteristics given the predisposition to the influence by nearby sources; for example, sea

spray in coastal environment or traffic-related emissions in roadside environment. Hong Kong has been operating a PM$_{10}$ monitoring network since 1998, which consists of six general stations and one roadside station. The network collects 24 h samples on quartz fiber filters on a 1 in 6 days schedule by high-volume (HV) samplers, which operate at a flow rate of 1.13 m$^3$ min$^{-1}$. The HV quartz fiber filters are used for gravimetric analysis and chemical speciation including major ions, elements, organic carbon (OC), and elemental carbon (EC) (Zhang et al., 2018). The PM$_{2.5}$ speciation network in Hong Kong started to

operate in 2011. PM$_{2.5}$ samples are collected on Teflon filters and quartz fiber filters by middle-volume samplers which operate at a flow rate of 16.7 L min$^{-1}$. The Teflon filters are used for gravimetric and elemental analyses while the quartz fiber filters are analyzed for major ions, OC and EC (Yu and Zhang, 2018). It should be noted that Si and Ti, which are important markers for quantifying dust contribution, are not determined in PM$_{10}$ samples due to the high background in ICP-OES analysis. On the other hand, the PM$_{2.5}$ network employs X-ray fluorescence technique for elemental analysis, and thus has no difficulty in

reporting the concentrations of these two elements. Additionally, carbonaceous components in PM$_{10}$ and PM$_{2.5}$ are determined using different thermal methods (NIOSH protocol for PM$_{10}$ and IMPROVE protocol for PM$_{2.5}$). In view of the aforementioned, the two PM monitoring networks in Hong Kong adopt different sampling and laboratory analysis protocols which would introduce uncertainties to the analysis results. The possibility of deriving a solid understanding of the composition and sources of PM$_{coarse}$ using existing data sets certainly requires further investigation.

We present in this work the first joint PM$_{10}$ and PM$_{2.5}$ speciation effort in Hong Kong in which all the sampling and chemical analysis work were conducted using identical methods and by the same laboratory. The aim is to obtain high quality composition data for PM$_{coarse}$. It has been reported in a number of studies that a notable fraction of PM$_{coarse}$ was often unable to be identified. Cheung et al. (2011) reported an up to 25 % contribution from such unidentified mass in Los Angeles area, while Putaud et al. (2010) reported 6–43 % in urban Europe. Although it has been suggested that the unidentified mass was

associated with liquid water content and mineral components, their exact contributions have remained largely uncharacterized. By using positive matrix factorization (PMF), we showed that the unidentified masses can be allocated to the resolved sources, providing qualitative and quantitative information on their origins. We propose the unidentified mass in PM$_{coarse}$ in our study region is mainly composed of unmeasured mineral components and liquid water content. The measured PM$_{coarse}$ in its entirety was successfully apportioned to various contributing sources by PMF, and the potential source origins are identified using

backward air mass trajectory analysis. With the robust source apportionment analysis, we found that fugitive dust associated with regional influence is the dominant contributor of high PM$_{coarse}$ loading in Hong Kong. The methodology and results from this study can serve to provide guidance to other locations with similar monitoring needs.



## 2 Methods

### 2.1 Ambient sampling

Aerosol sampling was conducted in Hong Kong at the Tuen Mun Air Quality Monitoring Station (TMC AQMS), which is located on the rooftop of a public library building (22°23'28.4" N, 113°58'37.1" E, ~30 m above ground level). The AQMS is situated in the northwestern part of the Hong Kong. The city, with a territory area of ~1110 km$^2$ and a population of ~7.5 million, is part of the bigger economic and business hub, the Greater Bay Area (GBA) (~56,000 km$^2$, population of ~85 million), in Guangdong province of China. Located in the sub-tropical region along the southeast coast of China, Hong Kong

exhibits a season-dependent air pollution characteristics that is closely related to the seasonal evolution of the East Asian Monsoon system. Generally, air pollution during colder seasons is more severe than in warm seasons. This will be elaborated when the measurement results are discussed.

Twenty-four-hour samples (midnight to midnight) for $PM_{10}$ and $PM_{2.5}$ were collected simultaneously on a once every three days schedule. The sampling lasted for over a year from 18 January 2020 to 9 February 2021. In each sampling event, one 47-

mm Teflon and one 47-mm quartz fiber filter samples were collected for each of the PM size fractions. The sample collection was accomplished by deploying two pairs of federal reference method samplers operated at a flow rate of 16.7 L min$^{-1}$. The first pair (Partisol Plus 2025, Thermo Fisher Scientific, MA, USA) were equipped with $PM_{10}$ sampling inlets to collect $PM_{10}$, whereas in the second pair (BGI PQ200, Mesa Labs, CO, USA) the Very Sharp Cut Cyclones were installed downstream of the $PM_{10}$ inlets for $PM_{2.5}$ fine particles collection. Field blanks (Teflon and quartz) were collected during the last sampling of

each month. All the filter samples were delivered back to the balance laboratory for conditioning followed by gravimetric analysis within one week. The filters were subsequently stored at –20ºC until chemical analysis.

### 2.2 Mass and chemical composition determination for $PM_{coarse}$

The mass concentration and chemical composition of $PM_{coarse}$ are determined as the difference between $PM_{10}$ and $PM_{2.5}$ measurements. The $PM_{10}$ and $PM_{2.5}$ samples were speciated using the identical protocol that has been adopted in the Hong

Kong $PM_{2.5}$ speciation network for regular monitoring of $PM_{2.5}$ composition since 2011 (Huang et al., 2014). The protocol is based on the speciation guideline by the U.S. Environmental Protection Agency (Chow and Watson, 1998). The design of joint sampling and chemical analysis of $PM_{10}$ and $PM_{2.5}$ eliminates data incompatible issues observed for data from the existing networks.

All the gravimetric and chemical analyses of the filter samples were conducted by the same laboratory in the Hong Kong

University of Science and Technology. PM mass concentration was determined on the Teflon filter samples by gravimetry with a digital microbalance (Sartorius AG, Model MC 5-0CE, Göttingen, Germany, sensitivity of ±1 μg) under a temperature- and relative humidity-controlled environment (20–23 °C and 30–40 %). Elements from Al to U were quantified on the Teflon filters by an energy dispersive X-ray fluorescence spectrometer (ED-XRF) (Epsilon 5, PANalytical, The Netherlands). OC and



EC were quantified on the quartz fiber filters with an aerosol carbon analyzer (DRI Model 2001A, Atmoslytic, Calabasas, CA,
USA) based on the thermal/optical reflectance method, adopting the IMPROVE_A temperature protocol (Chow et al., 2007).
Ionic species including $Cl^-$, $NO_3^-$, $SO_4^{2-}$, $NH_4^+$, $Na^+$, $Mg^{2+}$, $K^+$ and $Ca^{2+}$ were analyzed on the quartz fiber filters by ion
chromatography (IC) (Dionex ICS-1100, Thermo Fisher Scientific, MA, USA).

The species concentrations in $PM_{10}$ and $PM_{2.5}$ samples were blank corrected. The measurement precisions were propagated
from the precisions of volumetric measurements during sampling, chemical analyses, and field blank variability (Yu and
Zhang, 2018). Duplicate analysis of the aerosol samples was performed for every 10 measurements to derive precisions for
the chemical analyses. The measurement precisions for $PM_{coarse}$ speciation were propagated from the precisions of the $PM_{10}$
and $PM_{2.5}$ measurements.

### 2.3 Source apportionment by positive matrix factorization

Source identification and quantification for $PM_{coarse}$ was conducted by analyzing the speciation data matrix with positive matrix
factorization (PMF). PMF decomposes the speciation data matrix into factor profiles and factor contributions matrices with
non-negative constraints, with the objective of minimizing the uncertainty weighted differences between observed and
apportioned species concentrations represented by an objective function Q (Paatero and Tapper, 1994). The USEPA PMF 5.0
software was used for this undertaking (Norris et al., 2014). The fitting species include total $PM_{coarse}$ mass and a suite of
chemical species including $Na^+$, $NH_4^+$, $Mg^{2+}$, $Cl^-$, $NO_3^-$, $SO_4^{2-}$, OC, EC, Al, Si, K, Ca, Ti, V, Mn, Fe, Ni, Cu, Zn, and Pb. The
uncertainty of $PM_{coarse}$ mass was tripled to downweigh its influence in the source apportioning. This allows the total $PM_{coarse}$
mass to be apportioned mainly according to its covariance with other species. Concentrations below the method detection limit
(MDL) were replaced by $1/2 \times$ MDL with corresponding uncertainties set to be $5/6 \times$ MDL as recommended in the PMF user
manual. The input speciation data matrix consists of 123 $PM_{coarse}$ samples.

### 3 Results and discussion

### 3.1 Abundance and composition of $PM_{coarse}$

### 3.1.1 Annual average and comparison with other locations

The speciation data quality was evaluated by examining the consistency between species concentrations measured by different
methods; for example, gravimetric mass vs. mass from continuous monitor, gravimetric mass vs. reconstructed mass, $SO_4^{2-}$
vs. total S, and $K^+$ vs. total K, etc. Deming regression was applied in the examination using the Scatter Plot computer program
developed by Wu, which is available at https://doi.org/10.5281/zenodo.832417 (Wu and Yu, 2018). Details of the evaluation
are provided in Sect. S1 in the Supplement. In short, the evaluation shows the speciation data are of adequate quality for the
ensuing analyses.



The study-wide average concentration of $PM_{coarse}$ is 14.9±8.6 μg m$^{-3}$ (±standard deviation), accounting for 45 % of ambient $PM_{10}$ (32.9±18.5 μg m$^{-3}$). The daily concentrations range from 2.9 to 40.4 μg m$^{-3}$. The contribution of geological material is
estimated by assuming the crustal elements are in oxide forms, i.e., 1.89×[Al] + 2.14×[Si] + 1.2×[K] + 1.4×[Ca] + 1.67×[Ti] + 1.43×[Fe]. This component has the largest contribution, making up 5.2 μg m$^{-3}$ or 35 % of the $PM_{coarse}$ mass. The next important component is nitrate (2.2 μg m$^{-3}$, 15 %), followed by sea salt-related ions (i.e., $Na^+$, $Mg^{2+}$, and $Cl^-$) and organics (2×[OC]), which represent 11 % and 8 %, respectively. The composition forms a stark contrast with that of $PM_{2.5}$ (18.0±11.2 μg m$^{-3}$), in which carbonaceous components (1.6×[OC] and EC, 41 %) and secondary ions ($NH_4^+$, $NO_3^-$, and $SO_4^{2-}$, 38 %) are
the major components. The difference is consistent with combustion and secondary aerosol formation processes being the major sources of fine particles, whereas coarse particles are primarily generated by mechanical processes.

The annual average concentrations of $PM_{coarse}$ and selected major components measured in this study are compared with those in other locations in Table 1. Only studies that spanned at least one year or more and had all major species measured (i.e., elements, ions, OC and EC) are considered. Our $PM_{coarse}$ level is amid those in other urban locations, more than 2 times higher
than Milan in Italy and ~5 μg m$^{-3}$ higher than Central Los Angeles, and only half of that in Casa Grande in Arizona and a tenth of Lahore in Pakistan. Our concentration is also comparable to two roadside studies carried out in Bern in Switzerland and in London and Birmingham in the UK. We note the $PM_{coarse}$ concentration in a Hong Kong roadside study is ~10 μg m$^{-3}$ higher than the current study. Yet a straightforward urban vs. roadside comparison is not feasible given the roadside measurement was conducted more than 15 years ago. We also note that all the cited measurements were taken at least a decade ago. The
lack of more recent measurements highlights the need for more $PM_{coarse}$ speciation effort, considering the growing importance of $PM_{coarse}$ in aerosol mass loading and health effect contributions as $PM_{2.5}$ has been controlled effectively in many locations. Our $PM_{coarse}$ concentration is also 3–4 times lower than that measured in desert area in Arizona but one-third higher than a desert-like area in Lancaster in Los Angeles.

Geological material is the single largest component in $PM_{coarse}$ across all studies including ours, accounting for roughly 30–50
% (Lahore shows 74 %), underlining the importance in identifying fugitive dust sources (e.g., natural vs. anthropogenic) for effective mitigation of $PM_{coarse}$. We note that our nitrate concentration is the highest among all studies (except for the Lahore study, which is comparable to ours), constituting 2.2 μg m$^{-3}$ or 15 % of the $PM_{coarse}$. Coarse mode nitrate mainly forms by the uptake of $HNO_3$ by pre-existing alkaline particles forming $NaNO_3$ in reaction with sea salt and $Ca(NO_3)_2$ with soil dust. Our total carbon level of 0.7 μg C m$^{-3}$ is among the lowest compared to other studies, with 86 % of it coming from OC. A quarter
of $PM_{coarse}$ mass is regarded as unidentified in this study. The percentage share is among those observed in other studies, which range between 8 % and 38 %. The nature of the unidentified mass will be discussed in Sect. 3.2.2.

### 3.1.2 Seasonal variations in $PM_{coarse}$ mass and composition

The seasonal evolution of weather in Hong Kong is largely driven by the East Asian Monsoon system. Correspondingly, the atmospheric PM pollution in Hong Kong displays a distinct seasonal characteristic. In general, the PM loading in summer is





mainly governed by local emissions due to the prevailing southerlies carrying clean marine air mass. In winter, the prevailing northerlies place Hong Kong under the immediate downwind of the continental region with intense industrial and agricultural activities. Under this situation, the PM loading is affected by both local and regional sources. The transient seasons – spring and fall – have more mixed wind directions. The seasonal contrast in precipitation frequency and ambient temperature, both being higher in summer and lower in winter, also contributes to the variation in PM concentration across different seasons

(Louie et al., 2005; Yu et al., 2004).

    The sampling period in this study is divided into four seasons based on the observed meteorological and weather patterns as analyzed in Sect. S2 in the Supplement. Table 2 lists the starting and ending dates of individual seasons, along with the seasonal averages of PM concentrations and several meteorological parameters. Note that the two winter periods at the beginning and the end of the sampling program are regarded as two different winter periods considering the variability in weather conditions

and that they span mostly different calendar months.

    Figure 1 presents the $PM_{coarse}$ concentration and composition by season. The $PM_{coarse}$ exhibits a significant variation across different seasons, ranging from the lowest 8.1 µg m$^{-3}$ in summer to the highest 24.8 µg m$^{-3}$ in second winter. Washout by precipitation plausibly play a role in the seasonal contrast, given that summer takes up 75 % of the rainfall for the whole study period (Table 2). Mixing layer height appears to play an insignificant role in controlling the variation in $PM_{coarse}$ level. For

example, although the mixing height in the first winter is the lowest among all seasons (509 ± 402 m) while that in the second winter is the highest (874 ± 408 m), the $PM_{coarse}$ in the latter is more than twice higher than the former. The wind speed also shows small variation across the seasons, with a range of 1.9 to 2.3 m s$^{-1}$. This range corresponds to a Beaufort Scale Number of 1–2, referring to the light wind condition. The meteorological data imply changes in emission pattern and/or air mass origin are likely responsible for the seasonal variation in $PM_{coarse}$ levels.

The composition information indicates that geological material is largely responsible for the variability in $PM_{coarse}$. This component takes up 22–43 % of the $PM_{coarse}$ mass. The seasonal contrast in the contribution of this component could be attributed to enhanced wet deposition in warmer season and elevated contribution from regional transport in colder season. The unidentified mass also represents a major component in most seasons (except spring), accounting for 20–32 % of $PM_{coarse}$ mass. Like geological material, this fraction has a significantly enhanced contribution in the colder season compared to the

warmer season. As for other components, nitrate has the highest absolute contribution in spring and lowest in summer (3.2 vs. 1.2 µg m$^{-3}$). Organics are the highest in the second winter and lowest in the first winter, showing an order of magnitude difference (2.5 vs. 0.2 µg m$^{-3}$). The concentrations of sea salt-related ions (i.e., $Na^+$, $Mg^{2+}$, and $Cl^-$) are higher in the warmer season than that in colder season, which is consistent with the enhanced influence of marine air mass in the warmer season.



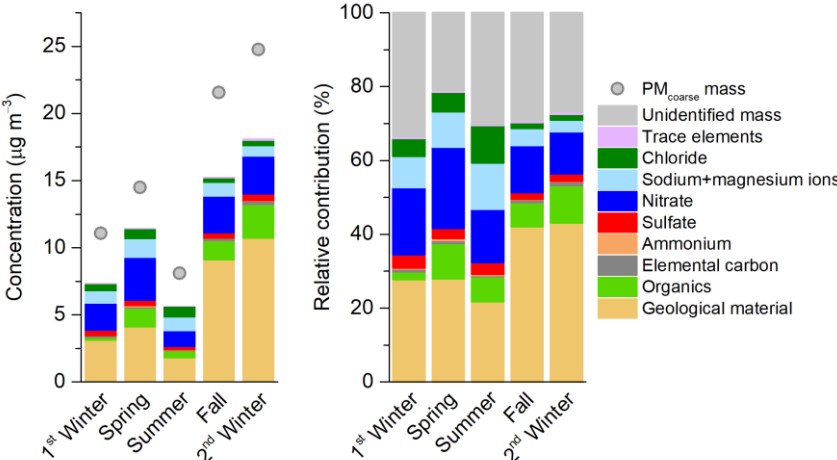

**Figure 1. Seasonal variations in concentration and composition of PM$_{coarse}$ observed at the Tuen Mun Air Quality Monitoring Station in Hong Kong. The left and right panels show the results in absolute concentration and relative contribution, respectively.**

**Table 1. Comparison of PM$_{coarse}$ concentration and major composition in microgram per cubic meter (percentage contribution to PM$_{coarse}$ shown in parentheses) in Hong Kong and measurements in other locations**

| Location | Measurement period | Number of measurements | PM$_{coarse}$ | Geological material | Nitrate | Total carbon | Unidentified mass | Investigator |
|---|---|---|---|---|---|---|---|---|
| Urban | | | | | | | | |
| Hong Kong | Jan. 2020–Feb. 2021 | 123 | 14.9 | 5.2 (35)[a] | 2.2 (15) | 0.7 (5) | 4.1 (26) | This study |
| Milan, Italy | Dec. 2009–Nov. 2010 | ~50 | 6.8 | 2.2 (32)[b] | <0.9 (13)[d] | 0.7 (10) | 2.6 (38) | Daher et al., 2012 |
| Central Los Angeles | Apr. 2008–Mar. 2009 | ~50 | 10.1 | 2.3 (23)[b] | 1.9 (19) | 1.1 (11) | 1.9 (18) | Cheung et al., 2011 |
| Casa Grande, Arizona | Feb. 2009–Feb. 2010 | ~60 | 30.6 | 16.4 (54)[b] | 0.7 (2) | 1.9 (6) | 7.4 (24) | Clements et al., 2014 |
| Lahore, Pakistan | Jan. 2007–Jan. 2008 | 63 | 142 | 105 (74)[b] | 2.4 (2) | 7.5 (5) | 24.1 (17) | Stone et al., 2010 |
| Roadside | | | | | | | | |
| London and Birmingham | Apr. 2000–Jan. 2002 | 101 | 12.4 | 4.7 (38)[c] | 1.4 (11) | 2.1 (17) | 0.9 (8) | Harrison et al., 2004 |
| Bern, Switzerland | Apr. 1998–Mar. 1999 | 76 | 19.6 | 4.9 (25)[b] | 1.1 (6) | 3.7 (19) | 4.4 (23) | Hueglin et al., 2005 |
| Hong Kong | Oct. 2004–Sep. 2005 | 40 | 25.9 | 7.3 (28)[a] | 1.9 (7) | 3.8 (15) | 6.7 (26) | Cheng et al., 2015 |
| Desert | | | | | | | | |
| Lancaster, Los Angeles | Apr. 2008–Mar. 2009 | ~50 | 9.4 | 3.6 (38)[b] | 0.5 (5) | 0.6 (6) | 3.4 (36) | Cheung et al., 2011 |
| Pinal County, Arizona | Feb. 2009–Feb. 2010 | ~60 | 45.5 | 23.5 (52)[b] | 0.8 (2) | 2.1 (5) | 13.6 (30) | Clements et al., 2014 |
| Cowtown, Arizona | Feb. 2009–Feb. 2010 | ~60 | 66.6 | 31.1 (47)[b] | 0.8 (1) | 8.6 (13) | 11.3 (17) | Clements et al., 2014 |

[a] Estimated by the investigators assuming oxides form of crustal elements.
[b] Estimated by the investigators assuming [Si] = 3.4×[Al] since Si was not measured.



<sup>c</sup> Estimated by the investigators using Ca and Fe as the markers for gypsum and soil dust, respectively.

<sup>d</sup> Only aggregate ions concentration was reported by the investigators.

**Table 2. Summary of season division, PM concentrations, and meteorological parameters in Tuen Mun during the sampling period**

| Season | Period | Number of aerosol samples | PM$_{coarse}$ (µg m$^{-3}$) | PM$_{2.5}$ (µg m$^{-3}$) | Temperature (°C) | Relative humidity (%) | Wind speed (m s$^{-1}$) | Total precipitation (mm) | Mixing height (m) |
|---|---|---|---|---|---|---|---|---|---|
| First winter | 18 Jan.–9 Mar. 2020 | 16 | 11.1 | 16.7 | 18.7±3.7 | 76±14 | 1.9±1.3 | 29.2 | 509±402 |
| Spring | 10 Mar.–17 May 2020 | 23 | 14.5 | 19.2 | 23.1±3.6 | 81±13 | 2.1±1.3 | 72.1 | 742±467 |
| Summer | 18 May–7 Oct. 2020 | 42 | 8.1 | 9.5 | 28.1±2.0 | 82±10 | 2.3±1.3 | 315.7 | 837±363 |
| Fall | 8 Oct.–28 Nov. 2020 | 18 | 21.6 | 22.3 | 23.5±2.5 | 67±14 | 2.2±1.2 | 1.5 | 870±425 |
| Second winter | 29 Nov. 2020–9 Feb. 2021 | 24 | 24.8 | 29.5 | 16.4±3.8 | 60±17 | 2.3±1.6 | 0.0 | 874±408 |

**3.2 Source characteristics of PM$_{coarse}$**

**3.2.1 Source identification by PMF analysis**

For the source apportionment analysis by PMF, the 4-factor solution was determined to be optimal by examining the mathematical outputs and physical interpretability of the resolved factor profiles in individual PMF solutions. Details of the examination are provided in Sect. S3 in the Supplement. The factor profiles are shown in Fig. 2. The first factor is clearly

associated with fugitive dust, as indicated by the high abundance of crustal elements (e.g., Al, Si, Ca, Ti, and Fe). The elemental ratios (e.g., Al/Si, Ca/Si, and Fe/Si) of this profile are similar to those of the local paved road dust samples reported by Ho et al. (2003). However, the ratios could also be a result of mixing of different dust types. The presence of carbonaceous components is suggestive of deposition of vehicular exhaust on road dust, and OC can also be linked to biological components in soil or vegetative debris emissions. Zn can be associated with tire wear (Pant and Harrison, 2013) or metallurgical process.

Given the various characters this profile possesses, this factor is named soil dust.

The second factor is to a certain degree similar to the first factor. It contains notable amount of OC, EC, Ca, Fe and Zn, suggesting it is also a dust-related source. The main difference is that this factor is depleted in Al, Si, and K and contains a high loading of Cu. Cu was reported to be a marker for brake wear, which is generated from the abrasion of brake lining material and brake discs (Pant and Harrison, 2013). Cu can also be associated with metallurgical process. Another characteristic

element in this factor is Ca. This element is enriched in construction dust because of the use of cementitious materials. The presence of OC and EC again points to the possible presence of road traffic and/or biological aerosols. For the lack of a better alternative name, the second factor is termed Cu-rich dust based on its characteristic Cu peak.

The third factor is marked by the high loading of Cl$^-$ with additional presence of Na$^+$ and Mg$^{2+}$, which are strong indicators for fresh sea salt. The molar equivalent of Cl$^-$ is balanced by that of Na$^+$ and Mg$^{2+}$, and it has an anion-to-cation equivalence





ratio of 0.99, adding credence to the validity of this factor. The last factor is loaded with a substantial fraction of Na$^+$ and Mg$^{2+}$,
which are markers for sea salt. The absence of Cl$^-$ and presence of NO$_3^-$ indicate this factor specifically represents aged sea
salt, given that Cl$^-$ in sea salt is actively depleted by gaseous HNO$_3$ forming nonvolatile NaNO$_3$. This factor is termed aged
sea salt mixed with secondary inorganic aerosols.

The stability of the PMF solution has been tested against the bootstrapping and displacement functions embedded in the PMF
software. The results show that the PMF solution is statistically robust for source analysis. Details of the uncertainty estimation
are summarized in Table S2 in the Supplement.

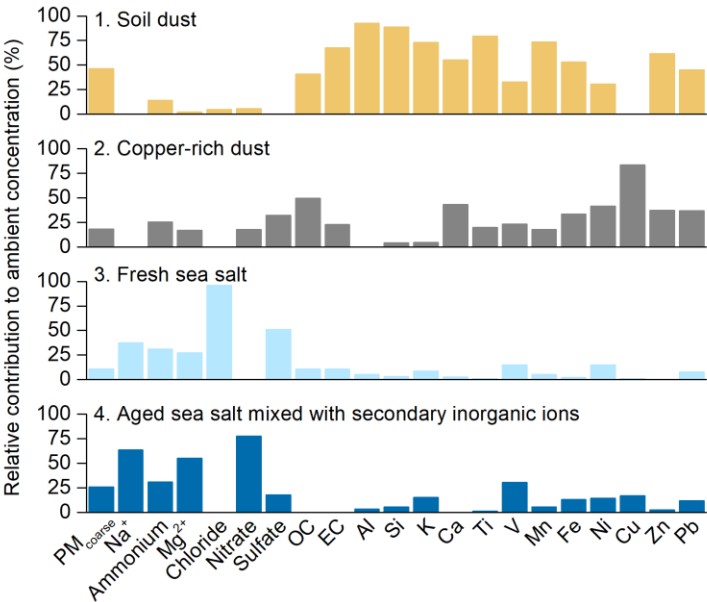

**Figure 2. Factor profiles resolved by positive matrix factorization for source apportionment of PM$_{coarse}$ measured at Tuen Mun Air
Quality Monitoring Station in Hong Kong.**

**3.2.2 Characterization of the unidentified PM$_{coarse}$ mass**

The total PM$_{coarse}$ mass was incorporated in PMF modeling. The apportioned masses show an excellent agreement with
measurements, with R$^2$ value of 0.98 and slope of 1.04 (intercept = –0.57). A test run was performed to examine if including
the total mass would affect the source apportioning. It shows that inclusion of total mass has a negligible impact on the PMF
solution. Specifically, the apportioning of all individual species is unaffected after including PM$_{coarse}$ mass as a total variable
(see Table S1 in the Supplement). The test result implies that the PM$_{coarse}$ mass in its entirety can be explained by the resolved
sources. Based on this finding, the unidentified mass can be allocated to the individual sources by taking the difference between
the PMF-apportioned mass and reconstructed mass in individual factors.

The unidentified mass derived from PMF (average = 5.2 μg m$^{-3}$) shows reasonable agreement with that from direct subtraction
using speciation data (average = 4.1 μg m$^{-3}$), with R$^2$ of 0.70 and slope of 1.07. Fugitive dust represents the largest contributor





to the unidentified mass, contributing 46 % (2.4 μg m$^{-3}$). The contribution by Cu-rich dust is 23 % (1.2 μg m$^{-3}$). Carbonate, a potentially important component in PM$_{coarse}$, is typically enriched with dust particles. As carbonate was not measured in this study, its quantity is estimated by two methods. The first method assumes all the excess cationic charge is balanced by carbonate. This method gives an average contribution of 0.6 μg m$^{-3}$. The second method assumes all Ca detected is in the form of CaCO$_3$. The resulting carbonate contribution is 1.5 μg m$^{-3}$ and is construed as the upper estimate. Considering Ca mostly

exists in the soil dust and Cu-rich dust factors, carbonate at most accounts for 42 % of the unidentified mass in the combined dust factors (3.6 μg m$^{-3}$), thus suggesting other unmeasured constituents exist.

A small amount of residue liquid water content (LWC) has been reported to be present in aerosol samples even at low relative humidity (RH) condition for gravimetric measurement. The thermodynamic equilibrium model ISORROPIA II (http://nenes.eas.gatech.edu/ISORROPIA) is applied to estimate the aerosol LWC under the RH and temperature conditions

of gravimetric measurement in the balance laboratory (i.e., temperature = 22 ℃, RH = 35 %) (Fountoukis and Nenes, 2007). The calculation is performed assuming an open system in which only aerosol phase concentrations are considered, and the aerosol is in metastable state. When comparing the LWC with individual soluble ions, including Na$^+$, Mg$^{2+}$, K$^+$, Ca$^{2+}$, Cl$^-$, nitrate and sulfate (shown in Fig. S5 in the Supplement), we find moderate to strong correlations between LWC and ions associated with sea salt: Na$^+$, Mg$^{2+}$, Cl$^-$, and nitrate (R$^2$ = 0.49–0.78). By contrast, sulfate, Ca$^{2+}$, and K$^+$ appear to be less

relevant (R$^2$ < 0.15). The results imply that sea salt components play a key role in governing the LWC in PM$_{coarse}$. The average LWC is estimated to be 1.2 μg m$^{-3}$, which agrees with the unidentified mass (1.6 μg m$^{-3}$) in the combined fresh and aged sea salt factors. The unidentified mass in aged sea salt mixed with secondary inorganic aerosols being higher than fresh sea salt (1.3 vs. 0.3 μg m$^{-3}$) is in line with the fact that NaNO$_3$ is more hygroscopic than NaCl.

After including carbonate and residue LWC, about half of the PMF-apportioned PM mass remains unidentified, and this

fraction is mainly contributed by the two dust-related factors. The mass discrepancy is likely attributed to the underprediction of geological mass in the mass reconstruction method, which only accounts for oxides of crustal elements. It is documented that other mineral constituents can exist in soil dust. For example, a field study in Morocco showed that over half of the PM$_{coarse}$ mass was made up of silicates (Kandler et al., 2009). Silicates commonly exist as illite and chloritoid, which contain mineral-bound water that is not considered in thermodynamic equilibrium model. Determining the missing components in the aerosol

dust and achieving a mass closure require further investigation with different techniques (e.g., microscopy). Overall, the results from the preliminary analysis of unidentified mass are consistent with the established knowledge. It provides support to the source apportionment results for the observed coarse particulates in its entirety, forming a strong basis for understanding their source origins.





## 3.3 Source contributions to PM$_{coarse}$

### 3.3.1 Seasonal variation


Figure 3 presents the absolute and relative source contributions by season in ascending order of PM$_{coarse}$ concentration. To better characterize the contribution by anthropogenic nitrogen oxides (NO$_x$) emission, the secondary nitrate is extracted from all the PMF-resolved factors. The two nitrate-free sea salt factors are grouped into one sea salt factor. In summer when PM$_{coarse}$ concentration is the lowest (8.1 µg m$^{-3}$), sea salt represents the largest contributor, contributing 47 % or 3.7 µg m$^{-3}$ of the ambient PM$_{coarse}$. Note that the source contribution is based on PMF-apportioned mass, thus the mass includes the contribution from residue LWC, which is mainly associated with enhanced uptake of water by aged sea salt aerosols. Soil dust represents the next important source, contributing 1.8 µg m$^{-3}$ or 24 %. Cu-rich dust contributes 1.0 µg m$^{-3}$ or 13 %. Secondary nitrate has a contribution of 1.2 µg m$^{-3}$ or 16 %.

PM$_{coarse}$ source composition changes gradually as PM$_{coarse}$ concentration increases from summer to the first winter and spring, and finally to fall and second winter. The trend indicates fugitive dust is the key driver for the elevated PM$_{coarse}$. In the fall and the second winter, soil dust contributed 12.9–15.2 µg m$^{-3}$ (58–61 %) to the ambient PM$_{coarse}$, whereas the Cu-rich dust contributed 3.2–4.6 µg m$^{-3}$ (14–18 %). The contribution by secondary nitrate is 2.4–2.7 µg m$^{-3}$, accounting for 9–12 %. This secondary component exhibits the lowest relative contribution compared to the 16–19 % contribution observed in other seasons.

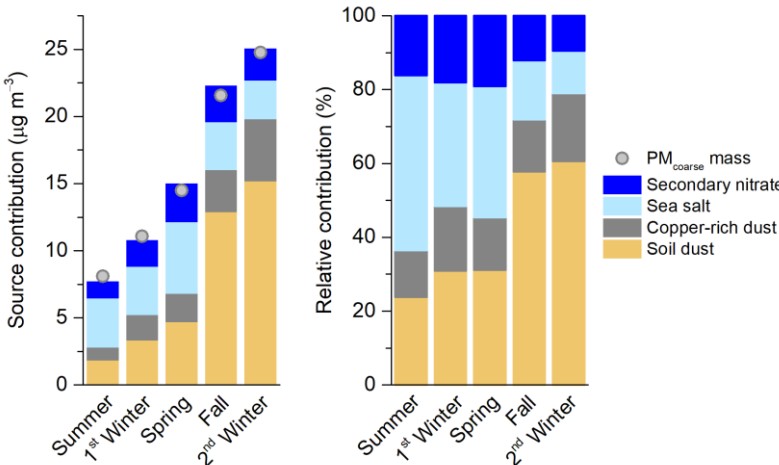


**Figure 3. Source contributions to PM$_{coarse}$ during the study period. The left figure shows the results in µg m$^{-3}$ while the right shows the results in percentage share. The circle markers on the left figure represent the PM$_{coarse}$ concentration measured by gravimetric analysis.**

### 3.3.2 Source contributions by air mass origins


The association between air mass origins and source influence was investigated through backward air mass trajectory analysis.

The back-trajectories were computed by the Hybrid Single-Particle Lagrangian Integrated Trajectory (HYSPLIT) model using





meteorological data from the 1° horizontal resolution Global Data Assimilation System (Stein et al., 2015). Past 48-hour back-trajectories of air mass reaching Hong Kong at 300 m height at the end of each sampling event at midnight were computed. The trajectories were clustered based on similarity between the trajectory endpoints. Four trajectory clusters are resolved, and

the mean for each cluster are displayed in Fig. 4. The average source contributions associated with each cluster is also shown in the same figure.

The source contributions of $PM_{coarse}$ associated with cluster 1 and 4 show contrasting features. Specifically, $PM_{coarse}$ concentration at the monitoring site is the highest when the site is influenced by cluster 1, reaching an average of 21.2 μg m$^{-3}$. During this period, the $PM_{coarse}$ is mostly contributed by dust-related sources, with soil dust and Cu-rich dust sources accounting

for 72 % and 18 %, respectively. Cluster 4 is under influence by marine air mass. The corresponding samples have an average $PM_{coarse}$ concentration of 8.0 μg m$^{-3}$, with sea salt being the largest contributor, accounting for 47 %. The two dust sources in total contributed to 38 % of the $PM_{coarse}$ mass. The total source contributions for cluster 2 and 3 are in between those of cluster 1 and 4, with source compositions reflecting the influence from the travelled source areas. By examining the individual trajectories in cluster 2 and 3, it can be seen cluster 2 is mostly composed of air masses passing through the coastal areas,

whereas cluster 3 consists of a mix of marine air masses from the east and short distance continental air masses from the northeast direction (see Fig. S6 in the Supplement). A common feature in cluster 2 and 3 is the relatively high contribution from secondary nitrate, which is 2 times or more higher than that in cluster 1 and 4 (2.5–3.0 μg m$^{-3}$ vs. 1.2–1.3 μg m$^{-3}$). Such a phenomenon could possibly be explained by the observation that cluster 2 and 3 have more mixed contributions from sea salt and $HNO_3$, whereas for the other two clusters either there is a deficiency in the availability of sea salt as in cluster 1 or

deficiency in $HNO_3$ as in cluster 4.



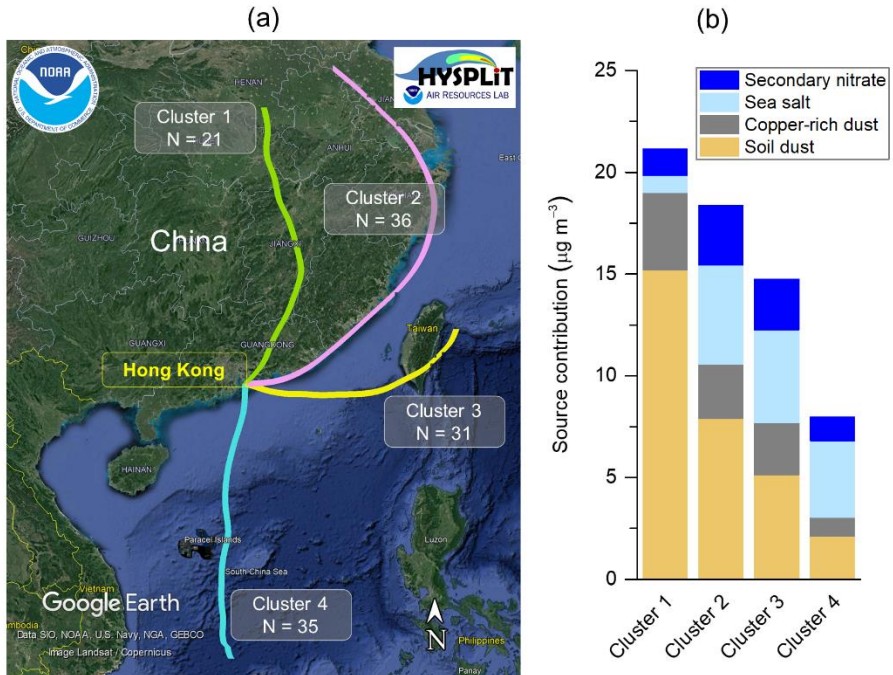

**Figure 4. Source contributions to PM_coarse grouped by air masses associated with different back-trajectory clusters. Past 48-hour backward trajectory of air mass reaching Hong Kong (height = 300 m above ground level) during the end of each sampling event at midnight are considered. Figure (a) shows the mean trajectories of the four clustered trajectories (Map data: © Google Earth,**
**Data SIO, NOAA, U.S. Navy, NGA, GEBCO, Image Landsat/Copernicus) while Fig. (b) shows the source contributions for the corresponding clusters.**

### 3.3.3 Potential source regions

The potential source areas are mapped by coupling the PMF-derived source contributions at the receptor with the associated backward air mass trajectory. In this analysis, the geographical domain of interest is divided and represented by a grid cell
matrix. By coupling the trajectory endpoints in the grid cells with the concentrations at the receptor, each grid cell will receive a value representing the potential source strength in the corresponding area. The Concentration-Weighted Trajectory (CWT) method is applied for the analysis (Hsu et al., 2003). In this method, each grid cell receives a weighted concentration value obtained by averaging the sample concentration that has associated trajectories crossing the corresponding grid cell, weighted by the residence time of air mass in that grid cell. The weighted concentration value (or CWT value) is expressed by Eq. (1):

$$CWT_{ij} = \frac{\sum_{l=1}^{L} C_l \tau_{ijl}}{\sum_{l=1}^{L} \tau_{ijl}}$$ (1)

where $C_l$ is the concentration at the receptor site associated with back-trajectory $l$, $\tau_{ijl}$ is the number of endpoints of trajectory $l$ falling into gird cell $i,j$ (i.e., the residence time of the trajectory in the grid cell), and $L$ is the total number of trajectories over a time period. To improve the robustness of the CWT analysis, the input trajectory information was augmented by considering all the trajectories calculated every three hours for each sampling day and assuming the same concentrations over the day (Petit





et al., 2017). The geographical domain was defined based on the spatial range of the trajectories traveled, with the dimension of grid cells set to be $0.5^o \times 0.5^o$. A weighting function was applied to down-weight grid cells with insufficient number of endpoints following the software guidelines. The CWT analysis was performed using the Zefir program (Petit et al., 2017). The analysis was performed by season to account for the potential variability in source strength and meteorological conditions. Figure 5 presents the CWT results for summer and the second winter and indicates the potential source areas. The results for

other seasons are displayed in Fig. S7 in the Supplement. It can be seen that for the soil dust and Cu-rich dust sources, the elevated contributions are associated with continental air masses originated from the north, whereas the sea salt-related contributions are associated with marine and coastal air masses. These results are consistent with the general understanding of source origins of these categories of sources. An important finding revealed from this analysis is that the GBA or the greater part of southern China is shown to have significant fugitive dust-related emission sources and that these dust sources are

implicated in causing days of high ambient $PM_{coarse}$ loading in Hong Kong.

Study on the fugitive dust sources in the related region is limited. A study featuring hourly measurements of trace elements in $PM_{coarse}$ and $PM_{2.5}$ coupled with PMF source apportionment analysis in Foshan (an industrial city in the GBA) resolved two dust factors, with the first being a mixture containing road dust, brake wear, and tire wear, and the second being construction-related dust (Zhou et al., 2018). The two dust factors in that study show similar features as those resolved in this study.

Specifically, their road dust/brake wear/tire wear factor accounted for over half of the coarse Al, Si, K, Ca, Ti and Fe by mass. The construction dust factor differs from the road dust/brake wear/tire wear factor by its higher abundance of Ca than Si. The enrichment in Ca is regarded as an indication of cementitious material commonly associated with construction activity. A point to note is that no Cu was measured by Zhou et al., hence it remains unclear to what extent the Cu-rich dust factor resolved in this study is similar to the construction dust factor. Zhou et al. reported that the $PM_{coarse}$ contributions by the road dust/brake

wear/tire wear and construction dust sources were 17.7 and 9.4 µg m$^{-3}$, respectively, during the seven-week monitoring in October–December 2014, higher than the 12.9–15.2 µg m$^{-3}$ and 3.2–4.6 µg m$^{-3}$ levels for the soil dust and Cu-rich dust factors during fall and second winter. This spatial gradient lends support to that the dust contributions in Hong Kong is associated with regional transport. Once entrained into the atmosphere, the lifetime of mineral dust can be up to several days and therefore it can be transported over long distance (over thousands of kilometers) and the concentration would decrease with transport

distance away from the source regions.





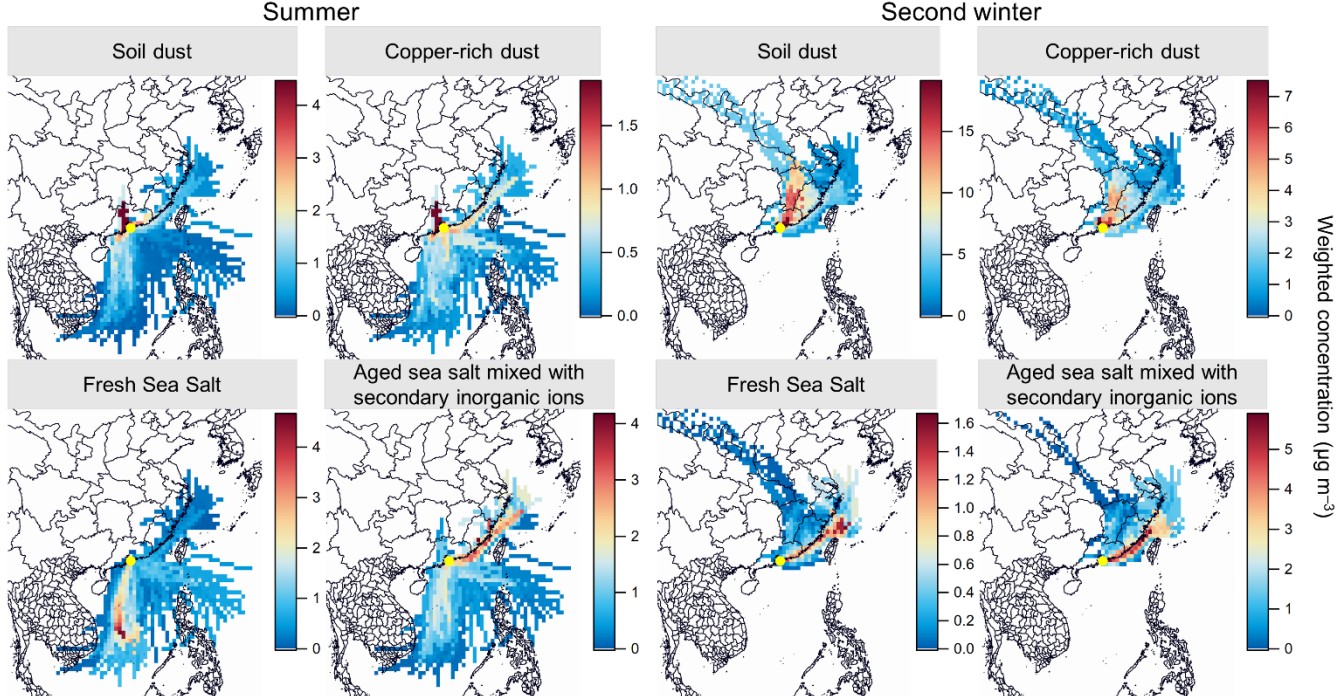

**Figure 5. Concentration-Weighted Trajectory results for individual PM$_{coarse}$ contributing sources in summer and the second winter. The location of the receptor site (Hong Kong) is represented by the yellow marker. The results for the other seasons are provided in Fig. S7 in the Supplement.**

**3.4 Implications to atmospheric research and public health**

As indicated in two field studies measuring size segregated PM composition in Hong Kong, the distribution of nitrate in fine and coarse mode particles in coastal environment depends on the amount of gaseous HNO$_3$ and alkaline particles (e.g., sea salt and soil dust) (Bian et al., 2014; Xue et al., 2014). The former is mainly controlled by the NH$_4$NO$_3$–NH$_3$ + HNO$_3$ equilibrium that is closely related to fine particles pH, temperature, and relative humidity, while the latter was shown to be more closely

related to sea salt. The source apportionment analysis for PM$_{coarse}$ in this study reaffirms sea salt plays a dominant role in the uptake of HNO$_3$ in our coastal environment. Based on the PMF results, 77 % of coarse nitrate is associated with sea salt, with the rest associated with fugitive dust. Despite the fact that fugitive dust-related aerosols represent a significant part of PM$_{coarse}$ loading in our study area, this component has a less important role to play in coarse nitrate formation. Nonetheless, the results indicate that controlling HNO$_3$ precursors would reduce nitrate in both PM$_{2.5}$ and PM$_{coarse}$. A limitation to note is that the

aerosol samples collected in this study were not corrected for sampling artifact of nitrate, which would affect the accuracy of the measured nitrate concentrations. The possible inter-particle interaction between fine and coarse particles on the PM$_{10}$ samples is also neglected, which potentially bias the nitrate measurements in the two size modes.



The comprehensive and high quality PM$_{coarse}$ speciation and source apportionment results identify fugitive dust as the significant contributor to PM$_{coarse}$, especially during high PM$_{coarse}$ days. It should be noted that the high loading of dust was not 400 caused by transient dust storm events, but occurred over the entire fall and winter season, indicating the constant emission of dust particles. A recent study conducted in northern China showed that coarse dust particles can act as a medium for rapid secondary inorganic and organic aerosols formation in highly polluted condition (Xu et al., 2020). Considering the southern China is more humid than northern China, our study region presents an atmospheric condition different from that in Xu et al.'s study, which is more favourable to adsorption of water on mineral dust, and consequently lead to different impacts on 405 atmospheric chemistry and climate (Tang et al., 2016). In this study, 90 % of coarse OC are apportioned to the two dust-related factors by PMF. Given both PM$_{2.5}$ and PM$_{coarse}$ in our study region typically experience long transport distance, more detailed speciation on organic markers might be helpful in elucidating the natural vs. anthropogenic and primary vs. secondary nature of the organics in PM$_{coarse}$.

Accumulative evidence has shown the positive link between adverse health effects and PM$_{coarse}$ exposure. Nationwide studies 410 in China have provided evidence for the association between short-term exposure to PM$_{coarse}$ and mortality and reduced pulmonary function in adult asthmatic patients (Chen et al., 2019; Lei et al., 2022). These studies indicate a stronger association in southern China compared to the northern part, which might be attributed to the difference in the source composition. For example, dust aerosols in the north typically contain higher proportion of windblown dust from natural sources while those in the south might have larger influence from industrial and traffic-related emissions. Oxidative potential of PM has been shown 415 to be a useful metric for PM health impact. Copper and humic-like substances (HULIS) are important active species in catalysing the formation of reactive oxygen species leading to oxidative stress in human body (Lin and Yu, 2011; Bates et al., 2019). The former is likely found in industrial emissions and non-tailpipe emissions (brake/tire wear) while the latter are likely associated with biological material in soil. In this study, the average concentrations of fine and coarse mode Cu are comparable, being 8.1±5.4 and 7.6±4.7 ng m$^{-3}$, respectively. Given that Cu is the important species governing the response of acellular 420 assay for PM oxidative potential measurement, the similar magnitude in concentration calls for further investigation into the sources and potential health effects of PM$_{coarse}$.

## 4 Conclusions

PM$_{coarse}$ has an important role to play in formulating policies to control PM$_{10}$ given its growing relative contribution to PM$_{10}$ loading in urban atmospheres. We have conducted the first joint chemical speciation of PM$_{10}$ and PM$_{2.5}$ in Hong Kong, a 425 coastal and highly urbanized city in southern China. This enables us to derive a high quality PM$_{coarse}$ composition data set spanning a 1 year long period from January 2020 to February 2021. The annual average concentration of PM$_{coarse}$ is 14.9±8.6 µg m$^{-3}$ (±standard deviation), representing nearly half (45 %) of ambient PM$_{10}$ (32.9±18.5 µg m$^{-3}$). The PM$_{coarse}$ also exhibit a





large seasonal variation, ranging from 8.1 µg m$^{-3}$ in summer to 24.8 µg m$^{-3}$ in the second winter period. Meteorological data suggest the seasonal contrast is driven by the variations in emission pattern and/or air mass origin.

Among the measured constituents, geological material calculated by assuming oxides of crustal elements represents the largest PM$_{coarse}$ component (35 %), followed by nitrate (15 %), sea salt ions (11 %) and organics (8 %). A quarter of PM$_{coarse}$ mass (4.1 µg m$^{-3}$) was regarded as unidentified mass according to a mass closure analysis. Positive matrix factorization analysis apportioned the PM$_{coarse}$ mass to four sources, including soil dust, Cu-rich dust, fresh sea salt, and aged sea salt mixed with secondary inorganic aerosols. Additionally, these four sources are able to account for the unidentified mass. The results show

that ~70 % of the unidentified mass is associated with the two dust factors, while the rest is residue liquid water content as implied from thermodynamic modeling using ISORROPIA II.

     The PM$_{coarse}$ concentration and corresponding source contributions show notable variations among samples influenced by different air mass origins. Specifically, the PM$_{coarse}$ concentration was averaged at 8.0 µg m$^{-3}$ when the site was influenced by marine air mass, with sea salt components being the largest contributor (47 %), followed by the two dust factors (38 % in

total). Significant elevation in PM$_{coarse}$ concentration was observed when the site was under the influences of air masses from the northern continental region, reaching 21.2 µg m$^{-3}$. The increase was largely driven by the enhanced contribution from the soil dust and Cu-rich dust factors, which contributed to 90 % of the PM mass in total.

     The source contribution and back-trajectory results were coupled and analyzed by the Concentration-Weighted Trajectory method to map the potential source areas. The results show that either the Greater Bay Area or the greater part of southern

China have a source intensity of fugitive dust-related emissions sufficiently large to result in the high ambient PM$_{coarse}$ loadings in Hong Kong, especially when the meteorological condition is favourable to regional transport of air pollutants. This study identified several aspects for further PM$_{coarse}$ or PM$_{10}$ research, including pinpointing the exact dust generation processes leading to the high PM$_{coarse}$ loadings in the study region, elucidating the roles of coarse particles in mediating secondary aerosol formation, and examining the potential health burden of PM$_{coarse}$ exposure through oxidative potential measurement.



*Data availability.* Chemical composition data presented in this study can be requested by emailing enquiry@epd.gov.hk or contacting the corresponding authors (envrykwong@ust.hk; jian.yu@ust.hk).

*Author contribution.* YKW, JZY and KKML formulated the overall design of the study. YKW, KML, and CY carried out the chemical analyses. YKW analyzed the data with contributions from JZY and KKML. YKW and JZY prepared the manuscript with contributions from all co-authors.

*Competing interests.* The authors declare that they have no conflict of interest.

*Disclaimer.* The content of this paper does not necessarily reflect the views and policies of the HKSAR Government, nor does mention of trade names or commercial products constitute an endorsement or recommendation of their use.

*Acknowledgements.* This work is supported by the Hong Kong Environmental Protection Department (HKEPD) (tender refs. 19-01121 and 19-01177). We thank Robert Tang and Rebecca Kwan of HKEPD for their inputs and assistance in project
logistics. We gratefully acknowledge the NOAA Air Resources Laboratory (ARL) for the provision of the HYSPLIT transport and dispersion model used in this publication.

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
