# Peer review of "Measurement report: Characterization and source apportionment of coarse particulate matter in Hong Kong: Insights into the constituents of unidentified mass and source origins in a coastal city in southern China"

_Atmospheric Chemistry and Physics, 2021_

## Author Comment (AC1)

*Response to Review Comments by Anonymous Referee #1 on "Measurement report: Characterization and source apportionment of coarse particulate matter in Hong Kong: Insights into the constituents of unidentified mass and source origins in a coastal city in southern China" by Yee Ka Wong et al.*

**General Comments by Anonymous Referee #1:**

The given manuscript discusses the importance of $PM_{coarse}$ in formulating policies due to its growing relative contribution to $PM_{10}$ loading in urban atmospheres. The paper uses the PM composition data derived from the measurements conducted in Hong Kong. The use of positive matrix factorization resulted in identifying four $PM_{coarse}$ sources, including soil dust, Cu-rich dust, fresh sea salt, and aged sea salt mixed +secondary inorganic aerosols. Results also showed that these four sources can explain unidentified fraction of $PM_{coarse}$. Overall, this work presents a simple approach to understand $PM_{coarse}$ composition/sources which can be applied to other locations with similar monitoring needs.

**Response to General Comments:**

We thank the reviewer for the comments and agreeing with the significance of this work. Our response to the comments is given in the following. The response text is marked in blue. References cited in this response document are placed at the end.

This work is nicely constructed, and my comments are listed below to consider:

- Please add reference to the sentence "For example, nitrate in the coarse mode is formed by the reaction between nitric acid (HNO3) from oxidation of NOx and pre-existing alkaline aerosols (e.g., sea salt and dust)."

    **Response:** We cited the study of Bian et al. (2014) here (Line 47–48), which used field measurements to show that the availability of sea salt and dust particles is one of the major factors affecting the formation of coarse nitrate.

- Please provide more details on the uncertainty matrix. I can't find any information on the uncertainty of species those concentration is above the detection limit.

    **Response:** The derivation of the precisions of measurements is given in the last paragraph of Sect. 2.2. These precision values were used as the PMF uncertainty matrix. We added the following statement in the corresponding section of the main text for clarification:

    Line 140–142: "The measurement precisions for each species in each sample described in Sect. 2.2 were used as the uncertainty inputs for the PMF modeling."

- Why was Deming regression applied?

    **Response:** Deming regression was applied because it considers the measurement uncertainties of both variables to be compared in the regression. This avoids biased fitting caused by only considering measurement error in the *y* variable as in ordinary least square regression. We added this explanation in the main text as follow:

    Line 152–153: "This technique is applied to consider the measurement uncertainties of both variables to be compared in the regression."

- Line 155: What is the basis for using a ratio of 2 to calculate organics, author should provide clarification.

    **Response:** Here we take the general understanding that coarse mode organics is more related to biological particles such as pollens, spores and vegetative detritus. These particles are enriched in more oxygenated compounds such as polyols and carboxylic acids. The ratio of 2 is a reasonable estimate adopted from the study

of Edgerton et al. (2009), in which the organic matter-to-OC ratio for coarse mode organics was reported. We included this justification in our main text as follow:

Line 161–163: "The coarse organics were estimated by multiplying the measured OC with a factor of 2, assuming the organics are mainly associated with biological particles, which are enriched in oxygenated compounds such as polyols and carboxylic acids (Edgerton et al., 2009)."

- Line 160: Add reference for using 1.6*OC.

  **Response:** We cited the study of Turpin and Lim (2001), which recommended the use of a ratio of 1.6 for typical urban aerosols.

- Add reference-"Coarse mode nitrate mainly forms by the uptake of HNO3 by pre-existing alkaline particles forming NaNO3 in reaction with sea salt and Ca(NO3)2 with soil dust."

  **Response:** We cited the study of Bian et al. (2014) to strengthen the statement.

- Line 220: Add reference for Si estimation.

  **Response:** The reference has been provided in the last column of Table 1.

- I am not convinced with the factor 1, it is a mix of two sources. Apart from crustal elements, there is also a significant contribution of Pb, V, Mn and Zn which suggests this source is not properly resolved.

  **Response:** We thank the reviewer for his/her views on the PMF analysis. We have considered the comments and revised the source identification section in Sect. 3.2. In the revised version of the manuscript, the back-trajectory analysis part is moved to the source identification section to aid the source interpretation.

  We also expanded the source identification part with additional discussion on the possible source categories for the dust factors. By considering the evidence from relevant studies, we renamed the first dust factor as soil dust/ industrial and coal combustion, and the second factor as construction dust/copper-rich emissions.

- The discussion about the seasonal contribution/variation of PMF factors should be enhanced. Currently, the given discussion is not sufficient to understand their origin. In addition, it would be great if author can also provide some insight on sources based on the previous receptor modeling results. Are the present results aligned with the previous observations?

  **Response:** We have revised the discussion on the seasonal variation in source composition, please be referred to Sect. 3.4.1.

  We also evaluated our source apportionment results with the useful information taken from a previous local $PM_{10}$ study by Yuan et al. (2013). The study analyzed the speciation data obtained from the local $PM_{10}$ network over an 11 year long period. In particular relevancy, the authors observed that the dust contributions were similar across different monitoring stations in Hong Kong, indicative of the regional nature of this source. The source origin of fugitive dust identified in our study aligns with that proposed by Yuan et al.

- Is the contribution of Cu-rich dust and construction dust by Zhou et al., comparable? What is the similarity between these two sites? Are the air masses originates from construction active area?

  **Response:** The Cu-rich dust factor is to certain extent similar to the construction dust by Zhou et al. Both factors show higher abundance of Ca than Si. Also, both factors are depleted in Al, Si, and K. However, no coarse mode Cu was reported in the PMF factor profiles by Zhou et al., and hence it remains uncertain to what extent the construction dust factor is similar to ours.

  The discussion on this factor has been expanded in the revised manuscript (Sect. 3.2.2). Based on the updated analysis, this factor is renamed as construction dust/copper-rich emissions.

Both Foshan (Zhou et al.'s study area) and Hong Kong are located within the Greater Bay Area (GBA). The Foshan city is one of the most important industrial hubs in the GBA, whereas Hong Kong is a commercial city with much less intense industrial activities. It is well documented that the air quality in Hong Kong is heavily influenced by emissions in the GBA, especially under meteorological conditions favorable to the transport of air pollutants. Therefore, the Foshan city might represent one of source areas responsible for the degraded air quality in Hong Kong imposed by regional transport of air pollutants.

As to the source areas that the continental air masses typically travelled before reaching Hong Kong, it is believed the air masses might pass through the various economic and industrial hubs in GBA or even the larger southern China region, where emissions from multiple sources are carried and mixed. Hence it is difficult to determine whether the air masses originate specifically from construction active areas.

- In the section 3.4, author mentioned that the aerosol samples were not corrected for sampling artifact of nitrate. Did author try to apply the correction and observed any change in the nitrate measurement?

  **Response:** We did not try to correct for the sampling artifact of nitrate. In principle, the artifact effect could be evaluated by performing co-sampling of aerosols with a denuder installed upstream of the filter to remove gas phase nitric acid. The extent of the nitrate sampling artifact is expected to be temperature-dependent and aerosol chemical composition-dependent, therefore varies from day-to-day. This variable nature makes its correction difficult. The effect of this type of artifact on coarse nitrate measurement warrants further investigation.

**References**

Bian, Q. J., Huang, X. H. H., and Yu, J. Z.: One-year observations of size distribution characteristics of major aerosol constituents at a coastal receptor site in Hong Kong – Part 1: Inorganic ions and oxalate, Atmos. Chem. Phys., 14, 9013–9027, https://doi.org/10.5194/acp-14-9013-2014, 2014.

Edgerton, E. S., Casuccio, G. S., Saylor, R. D., Lersch, T. L., Hartsell, B. E., Jansen, J. J., and Hansen, D. A.: Measurements of OC and EC in coarse particulate matter in the Southeastern United States, J. Air Waste Manage., 59, 78–90, https://doi.org/10.3155/1047-3289.59.1.78, 2009.

Turpin, B. J. and Lim, H. J.: Species contributions to $PM_{2.5}$ mass concentrations: Revisiting common assumptions for estimating organic mass, Aerosol Sci. Technol., 35, 602–610, https://doi.org/10.1080/02786820119445, 2001.

Yuan, Z. B., Yadav, V., Turner, J. R., Louie, P. K. K., and Lau, A. K. H.: Long-term trends of ambient particulate matter emission source contributions and the accountability of control strategies in Hong Kong over 1998–2008, Atmos. Environ., 76, 21–31, http://dx.doi.org/10.1016/j.atmosenv.2012.09.026, 2013.

---

## Author Comment (AC2)

*Response to Review Comments by Anonymous Referee #2 on "Measurement report: Characterization and source apportionment of coarse particulate matter in Hong Kong: Insights into the constituents of unidentified mass and source origins in a coastal city in southern China" by Yee Ka Wong et al.*

**General Comments by Anonymous Referee #2:**

This study performed chemical speciation for $PM_{2.5}$ and $PM_{10}$ samples collected in Hong Kong during 2020/01-2021/02. The results showed that the annual average concentration of $PM_{coarse}$ ($PM_{10}$-$PM_{2.5}$ mass) accounted for ~50% of $PM_{10}$. Unlike $PM_{2.5}$, only ~75% of $PM_{coarse}$ mass was explained by identified chemical components. The authors supposed that the unidentified part was dominated by geological components and aerosol liquid water. Moreover, several tools were utilized to apportion $PM_{coarse}$ to specific sources and areas, particularly for the unidentified fraction. In general, this manuscript is well organized and written. But two major issues should be addressed before the consideration for publication.

**Response to General Comments:**

We thank the reviewer for the comments and appreciating our work. Our response to the comments is given in the following. The response text is marked in blue. References cited in this response document are placed at the end.

1.  In this work, the thermodynamic equilibrium model (ISORROPIA II) was adopted to estimate aerosol liquid water (ALW) in $PM_{coarse}$. After mass closure and PMF analysis, the authors concluded that the unidentified $PM_{coarse}$ (4.1 μg m$^{-3}$, ~25%) was substantially contributed by ALW (1.2 μg m$^{-3}$).

    Have the authors performed mass closure for $PM_{2.5}$ or $PM_{fine}$? Because the fine particles are more enriched with water soluble components (e.g., secondary inorganic ions), ALW should contribute more fractions to $PM_{2.5}$. According to section 3.1.1 (lines 158-160), it seems that $PM_{2.5}$ is mainly composed of $NH_4^+$, $NO_3^-$, $SO_4^{2-}$, OC, and EC (~80%).

    If ALW contributes a significant fraction of $PM_{coarse}$ based on filter sampling, there's no reason that it contributes less to $PM_{fine}$.

    In fact, ALW is not stable on filters, and is subject to loss during long-term sampling and transportation.

    So, the contribution of ALW to unidentified $PM_{coarse}$ might not be estimated appropriately with the current study design.

    **Response:** The reviewer might misunderstand the nature of LWC in the discussion, confusing LWC as existing under ambient condition vs. residue LWC held tightly onto particles under the dry weighing conditions, as stated in Line 331–333 of the updated manuscript file:

    "The thermodynamic equilibrium model ISORROPIA II (http://nenes.eas.gatech.edu/ISORROPIA) is applied to estimate the aerosol LWC under the RH and temperature conditions of gravimetric measurement in the balance laboratory (i.e., temperature = 22 °C, RH = 35 %) (Fountoukis and Nenes, 2007)."

    Still, we applied the ISORROPIA model with the same setting to the $PM_{2.5}$ composition data to estimate residue LWC bound to $PM_{2.5}$ under the dry weighing conditions (Fountoukis and Nenes, 2007). The amount of LWC was calculated to be negligible (average = 0.01 μg m$^{-3}$), suggesting the inorganic ions in $PM_{2.5}$ do not retain LWC as effective as sea salt in $PM_{coarse}$ under the dry weighing conditions.

    The lower residue LWC in $PM_{2.5}$ is supported by the better mass closure in $PM_{2.5}$ compared with $PM_{coarse}$. The average reconstructed-to-measured mass ratio for $PM_{2.5}$ is 0.90±0.08, better than the ratio of 0.72±0.10 for $PM_{coarse}$. The results align with the theoretical residue LWC in $PM_{2.5}$ being lower than that in $PM_{coarse}$.

2. When input $PM_{coarse}$ mass for PMF analysis, it was presumed that the unidentified $PM_{coarse}$ fraction have the same sources as identified components.

In this work, four factors linked with soil dust, copper-rich dust, fresh sea salt, and aged sea salt were identified using measured species data. Since understanding the sources and formation pathways of PM largely depends on how well they are identified, the sources of un-speciated coarse PM are unknown and might not be the same as measured species. If the unknown fraction of coarse PM was apportioned to the four identified factors, some factors contributions would be over-estimated. Because PMF may over-attributed $PM_{coarse}$ to certain factors as it fits measured species (Shrivastava et al., 2007). This will occur if makers for unknown $PM_{coarse}$ are not included in the PMF model (Shrivastava et al., 2007).

Therefore, the source apportionment method for unidentified $PM_{coarse}$ mass is not appropriate. The authors should focus on sources of identified $PM_{coarse}$ components.

**Response:** We thank the reviewer for sharing his/her critique on this, which we agree in some sense. But here we would like to raise an opinion that the issue of over-attributing a species in PMF modeling arises when the model is set to fit the species explicitly, and this typically applies to source marker species, and in some occasions to the bulk species being apportioned, such as OC in the study of Shrivastava et al. (2007). However, we took a different approach to apportion $PM_{coarse}$ in our PMF modeling. Specifically, the $PM_{coarse}$ concentration was set to be a total variable with the concentration uncertainties tripled to decrease their weight in the model fit. This would allow the $PM_{coarse}$ to be apportioned based on its temporal covariance with other input species, in other words, not being forced to fit to the apportioned factors, avoiding the issue of over-attribution.

To examine whether the $PM_{coarse}$ was apportioned in this specified way, the PMF solutions with and without considering $PM_{coarse}$ were compared, as documented in Sect. 3.3 of the revised manuscript. The two solutions are identical in multiple aspects including the chemical composition of the factor profiles resolved and the modeling performance of all individual species. The test demonstrated that the $PM_{coarse}$ was apportioned purely based on its covariance with other species without affecting the apportioning of other species. It is because if a source significant enough to affect the $PM_{coarse}$ variation is missing, and the model attempted to fit the $PM_{coarse}$, some of the included species would be compromised. The absence of influence by $PM_{coarse}$ and the excellent agreement between the apportioned and measured masses ($R^2 = 0.98$; slope = 1.04) are two important signs to indicate the temporal variation of $PM_{coarse}$ in its entirety, including the unidentified fraction, can be well captured by the resolved sources.

**References**

Fountoukis, C. and Nenes, A.: ISORROPIA II: A computationally efficient thermodynamic equilibrium model for $K^+$–$Ca^{2+}$–$Mg^{2+}$–$NH_4^+$–$Na^+$–$SO_4^{2-}$–$NO_3^-$–$Cl^-$–$H_2O$ aerosols, Atmos. Chem. Phys., 7, 4639–4659, https://doi.org/10.5194/acp-7-4639-2007, 2007.

Shrivastava, M. K., Subramanian, R., Rogge, W. F., and Robinson, A. L.: Sources of organic aerosol: Positive matrix factorization of molecular marker data and comparison of results from different source apportionment models, Atmos. Environ., 41, 9353–9369, https:// doi.org/10.1016/j.atmosenv.2007.09.016, 2007.

---

## Author Response (AR2)

Title: Characterization and source apportionment of coarse particulate matter in Hong Kong: Insights into the constituents of unidentified mass and source origins in a coastal city in southern China
Author(s): Yee Ka Wong et al.
MS No.: acp-2021-1030

*Point-by-Point Response to Editor's Comments*

Comments to the author:

The authors have reasonably well addressed the comments of the two anonymous referees and they have modified their manuscript accordingly. However, alterations and corrections are needed for both the Main text and Supplement before the manuscript can be published in ACP:

We thank the editor for the comments. Our response text is marked in blue in this document. The revised text in the main manuscript is also marked in blue.

Main text:

Line 14: Replace "of its" by "for its". **Response:** Corrected.

Line 35: Replace "play important" by "plays important". **Response:** Corrected.

Line 39: Replace "were inconclusive" by "was inconclusive". **Response:** Corrected.

Line 47: Replace "are mostly" by "is mostly". **Response:** Corrected.

Line 55: Replace "originated from" by "originating from". **Response:** Corrected.

Line 85: Replace "in Los" by "in the Los". **Response:** Corrected.

Line 99: Replace "of the Hong" by "of Hong". **Response:** Corrected.

Line 102: Replace "a season-dependent air pollution characteristics that is" by "season-dependent air pollution characteristics that are". **Response:** Corrected.

Line 109: Replace "were equipped" by "was equipped". **Response:** Corrected.

Lines 136-137: Replace "with positive matrix factorization (PMF)" by "with PMF"; the acronym PMF should not be defined again as this was already done in line 88. **Response:** Corrected.

Line 212: Replace "are likely" by "and are likely".

**Response:** The statement was revised as below:

"The meteorological data imply that the seasonal variation in PM$_{coarse}$ levels is likely caused by changes in source intensity and/or air mass origin."

Line 265: Replace "mean for" by "means for". **Response:** Corrected.

Line 273: Replace "cluster 2" by "clusters 2". **Response:** Corrected.

Line 294: Replace "is also" by "are also". **Response:** Corrected.

Line 302: Replace "while deplete" by "while the are depleted". **Response:** Corrected to "while they are depleted".

Line 350: Replace "in thermodynamic" by "in the thermodynamic". **Response:** Corrected.

Line 370: Replace "from continental" by "from the continental". **Response:** Corrected.

Line 383: Replace "is associated" by "are associated". **Response:** Corrected.

Line 400: Replace "gird cell" by "grid cell". **Response:** Corrected.

Line 420: Replace "in coastal" by "in a coastal". **Response:** Corrected.

Line 428: Replace "for sampling" by "for a sampling". **Response:** Corrected.

Line 430: Replace ", therefore" by "; therefore". **Response:** Corrected.

Line 438: Replace "the southern" by "that southern". **Response:** Corrected.

Line 440: Replace "lead to" by "leads to". **Response:** Corrected.

Line 452: Replace "in human" by "in the human". **Response:** Corrected.

Line 463: Replace "exhibit a" by "exhibits a". **Response:** Corrected.

Line 482: Replace "to regional" by "for regional". **Response:** Corrected.

Supplement:

Page S1, before line 1: A header with "Supplement", the manuscript title, authors and affiliations is needed.

**Response:** We noted the following statements from the Supplement section of the manuscript submission guideline (https://www.atmospheric-chemistry-and-physics.net/submission.html#manuscriptcomposition):

"Supplements will receive a title page added during the publication process including title ("Supplement of"), authors, and the correspondence email. Therefore, please avoid providing this information in the supplement."

This is why we did not include the title, authors, etc. Do let us know if the policy has changed and we are happy to make revision.

Page S5, line 2: Replace "factor was" by "factors was". **Response:** Corrected.

Page S5, line 15: Replace "Five-factor" by "The five-factor". **Response:** Corrected.

Page S7, line 1 of Figure caption: Replace "Cluster 1" by "Clusters 1". **Response:** Corrected.